# Variable-to-Variable Huffman Coding: Optimal and Greedy Approaches

**DOI:** 10.3390/e24101447

**Published:** 2022-10-11

**Authors:** Kun Tu, Dariusz Puchala

**Affiliations:** 1School of Mathematical Sciences, Yangzhou University, 88 South Daxue Road, Yangzhou 225002, China; 2Institute of Information Technology, Lodz University of Technology, 8 Politechniki Avenue, 93-590 Lodz, Poland

**Keywords:** Huffman coding, variable-to-variable m-gram entropy coding, lossless data compression

## Abstract

In this paper, we address the problem of *m*-gram entropy variable-to-variable coding, extending the classical Huffman algorithm to the case of coding *m*-element (i.e., *m*-grams) sequences of symbols taken from the stream of input data for m>1. We propose a procedure to enable the determination of the frequencies of the occurrence of *m*-grams in the input data; we formulate the optimal coding algorithm and estimate its computational complexity as $O(mn^{2})$, where *n* is the size of the input data. Since such complexity is high in terms of practical applications, we also propose an approximate approach with linear complexity, which is based on a greedy heuristic used in solving backpack problems. In order to verify the practical effectiveness of the proposed approximate approach, experiments involving different sets of input data were conducted. The experimental study shows that the results obtained with the approximate approach were, first, close to the optimal results and, second, better than the results obtained using the popular DEFLATE and PPM algorithms in the case of data that can be characterized by highly invariable and easy to estimate statistics.

## 1. Introduction

Today, as members of the information society, we extensively use computer systems operating with large amounts of data that must be transmitted over computer networks and archived in computer systems with the use of mass storage devices. We are also witnessing the constant increase in the capabilities of modern data acquisition devices and the continuous growth of the popularity of mobile electronic equipment, which places higher and higher demands on the amount of data required to be transmitted between various computer systems (e.g., in embedded systems including Internet of Things (IoT) appliances interconnected with 5G cellular networks). Bearing this in mind, we can expect these trends to increase regularly in the future. For this reason, it is very important to reduce the actual size of the data to be transmitted without the loss of information conveyed by the data itself. In the case of static images, video sequences, or audio recordings, we agree to some distortion in the reconstructed data. For example, minor modifications to the colors of pixels in a natural image do not mean the loss of information, because we are still able to read the pictured content. In such a scenario, we can speak about lossy data compression, which in practice can result in even a few hundred times reduction in the size of the input data (c.f. Joint Photographic Experts Group (JPEG) or Moving Picture Experts Group (MPEG) standards [1]). Conversely, the situation is different in relation to text data, executable files of computer programs, and topographic data including coordinates, geometric descriptions of terrain and field objects, and descriptions of movement trajectories of objects, etc. In these cases, any change in the input data is not allowed, which means that the data must be reconstructed without distortion. Consequently, we can speak of lossless compression. Compared to lossy compression, lossless compression not only places additional restrictions on the data compression methods but also limits the compression ratios that can be obtained.

Data compression methods are the techniques of data coding through which one message is represented in the form of another message using a different alphabet that can be characterized by a more compact representation (i.e., requiring a smaller number of bytes). Lossless data compression methods can be roughly grouped into the following three categories:Block-to-variable codes: this is a type of coding technique where a variable-length output codeword is assigned to each symbol of the input alphabet. The length of the codeword strictly depends on the frequency (probability) of the occurrence of the input symbol according to the following rule—the more probable a symbol the shorter the output code. The compression is obtained when frequently appearing symbols are coded in a more efficient way. The following coding techniques belong to this category: phased-in codes [2], redundancy feedback coding, Shannon–Fano codes [3,4], and Huffman coding [5,6].Variable-to-block codes: this is a category of coding techniques where we assign fixed-size output codewords to variable-length sequences of symbols from the input data. Compression is achieved by encoding long sequences with shorter fixed-length codes when long sequences of input symbols appear with high frequency in the input data stream. This category includes the following coding methods: Tunstall codes [7], digram coding [1], and the dictionary-based techniques LZ77 [8,9], LZ78, [10], and LZW [11].Variable-to-variable coding: this is a category of techniques that encode the variable-length input data sequences using variable-length output codewords. These techniques allow for higher compression ratios because they combine the advantages of both the abovementioned classes of methods. This category includes, for example, the modified LZW algorithm [12] (where the size of the output codewords is not constant but depends on the level at which the dictionary is filled with sequences of data), DEFLATE [13] (uses dictionary and Huffman coding, in which the dictionary indexes are coded with block-to-variable codes depending on their frequencies of occurrence), frequency-directed run-length (FDR) codes [14], variable-length input Huffman coding [15] (applicable to binary data where the sequences of 0’s are coded using the Huffman method), selective Huffman coding [16,17,18] (where only the most frequently occurring binary sequences are encoded with the Huffman method), and the dual-tree based approach [19,20] (the apt combination of Tunstall and Huffman coding, which improves the efficiency of the second one).

In this paper, we use Huffman coding. As mentioned above, Huffman coding is a block-to-variable coding method that depends on the known frequencies of symbols in the input stream. It assigns the shortest code to the most frequently occurring symbol. It is convenient to implement with a code in time linear to the number of input frequencies when the frequencies are sorted. See Section 2 for the details of the Huffman coding algorithm. In fact, it is a type of prefix code, and it is so widespread that “Huffman code” is widely viewed as a synonym for “prefix code”. There is no ambiguity when decoding the generated bitstream, and it ensures that the decoded information is lossless and absolutely precise. The original algorithm was proved to be optimal for a symbol-by-symbol coding with a known input probability distribution. Although there are many alternatives performing better than the classical Huffman coding in practice, Huffman coding remains indispensable. For example, it is widely applied to fax, text, GZIP, JPEG, PNG, MP3, etc. Its necessity is accepted when we are dealing with backup files containing important information. To promote the efficiency of Huffman coding, researchers have tried to combine a fixed number of symbols together (“blocking”) to increase the compression. As the size of the block increases, the variant-Huffman coding approaches the entropy limit. However, there are some theoretical difficulties. For instance, first, the time complexity of the algorithm becomes exponential with regard to the size of the block, which makes the approach impractical. Second, it is difficult to decide the value of the “fixed number”. The algorithm is not as flexible once the value is fixed, and the fixed number approach may be not compatible with the classic Huffman coding, i.e., a fixed m>1 is not guaranteed to be better than m=1. The theoretical problems introduce serious obstacles to the applications of the Huffman coding.

In this article, we study these problems in theory and move the Huffman coding a step forward, which may shed light on further future applications. Specifically, we address the problem of variable-to-variable coding using the natural extension of the classical Huffman coding to *m*-grams, i.e., the sequences of symbols for the values of *m* higher than 1. The novelty of this paper includes the following five points. (i) We provide a technique that enables determination of the frequencies of the occurrence of the symbol sequences (*m*-grams) in the input data with various lengths ranging from 1 to *m*. (ii) A formulation of the optimal *m*-gram Huffman data coding algorithm is proposed. Additionally, an estimation of its computational complexity is shown. (iii) In order to promote the efficiency of the algorithm, we propose an approximate approach based on a greedy heuristic. It can be characterized by linear computational complexity in relation to the size of the input data. (iv) The decoding algorithm is considered and shown. (v) To verify the effectiveness of the proposed approximate approach, experiments are conducted for different sets of input data.

## 2. Huffman Coding

Huffman coding is a popular technique for block-to-variable lossless compression of data (see [5]). In this technique, based on the frequency of occurrence, we assign to each symbol coming from the input data the individual binary codewords of variable length according to the following intuitive rule: the more frequent a symbol is, the shorter the codeword assigned to it. In order to create the codewords, we take advantage of binary trees that guarantee obtaining prefix codes, which denotes the class of codes where no codeword is a prefix of any other codeword. Codes having this property are highly demanded in variable-length coding because they enable unambiguous decoding of data even from a sequence of concatenated codewords. It should also be noted that Huffman coding constructs a set of variable-length codewords that can be characterized by the shortest average length [21].

In order to decode the data at the decoder, we need to know the codewords assigned to the individual symbols occurring in the input stream. The codewords (for all input data symbols) can be placed into a stream of compressed data, but it is usually more effective to transmit information about the frequency of the occurrence of the individual symbols. Based on this information, the decoder is able to reconstruct codewords by building and using the same binary tree. In some cases, it can be also advantageous to use fixed codewords, which are known to both the coder and decoder. It should be noted that such an approach can be highly practical in cases where the given class of input data is characterized by the stable frequency distribution of symbols. We demonstrate the Huffman coding technique through the following elementary example.

Let us assume that the input data stream is constructed over the alphabet of the form A={a,b,c} and that the following exemplary sequence of symbols I is to be coded, i.e., we have I={aaaabbaaaabbbbbbaabcabbccabbaaaaabbccaaaaa}. Obviously, we have three different symbols in A, which means that we need 2 bits to code each of the symbols when using natural binary coding. With the size of the input data n=|I|=42, there are 84 bits required to code the whole input sequence I. In the case of Huffman coding, in the first place, we have to evaluate the frequencies of the occurrence of all the input symbols in the alphabet A. This step does not consume much time (computational complexity O(n)) and only requires iterating the entire input data sequence I and counting the occurrences of the symbols. The results obtained for the exemplary sequence I are collected in the second column of Table 1.

The next step is to create a Huffman code tree and to determine the codewords assigned to individual symbols. We start constructing the tree with a forest composed of nodes representing the symbols from the alphabet together with their frequencies of occurrence (see Figure 1).

The whole process is iterative, and in each iteration leading to the Huffman code tree, we have to select two trees from the forest with the lowest frequencies of occurrence, and then combine them into one tree in which the root node represents the total number of occurrences of all the symbols in the newly formed tree (see Figure 2).

We repeat these operations until only one tree is left. The resulting tree is the complete Huffman code tree that we are looking for (see Figure 3).

Next, in order to retrieve the codewords for the individual symbols, we have to traverse the tree from the root node to each leaf. For each symbol, the path information from the root node to its node is stored in the form of concatenated bits, where ‘0’ describes the left and ‘1’ the right subtree (this assignment is fully arbitrary and can be inverted). The resulting path descriptors are the searched for codewords. For the considered exemplary case, the codewords obtained on the basis of the code tree from Figure 3 are collected in the third column of Table 1. It should be noted that the length of the codewords depends strictly on the height of the tree. The Huffman code tree has the smallest height when all the symbols have an equal probability of occurrence. When the frequencies of the occurrence of the following symbols form the elements of a Fibonacci sequence, the resulting tree is unbalanced, which results in its maximum height [21].

If we use the Huffman codewords from Table 1 to code the exemplary sequence of data I, then it is elementary to check that we can obtain the coded sequence of the following form C={11110000111100000000000011000110000010110000111110000010111111}. It is also simple to verify that the coded sequence requires 62 bits, which gives an average number of 1.476 bits per single symbol related to the total number of symbols n=42. This is more than a 26% improvement. In this place, we can ask how close this result is to the theoretical limit. Of course, the theoretical limit is determined by the first-order entropy calculated as [21]:(1)$$\begin{array}{c} H\;=\;-\sum_{i=0}^{|A|-1}p_{i}\log_{2}p_{i}, \end{array}$$
where $p_{i}$ for i=0,1,…,|A|−1 describe the probabilities of the occurrence of the individual symbols from the alphabet A. In our example, these probabilities can be calculated as $p_{i}\;=\;f_{i}/n$, with $f_{i}$ describing the frequencies of the occurrence of the symbols (these frequencies are in the second column of Table 1). Hence, for the symbol ‘a’ we have $p_{0}\;=\;22/42$, for the symbols ‘b’ and ‘c’ we have $p_{1}\;=\;15/42$ and $p_{2}\;=\;5/42$, respectively. Based on these results, one can easily calculate the entropy using the formula (Equation 1), which leads to the result of H=1.385 bits per symbol.

We can clearly see that the result of H=1.476 bits per symbol obtained with the Huffman coding was higher than the theoretical limit of H=1.385 bits per symbol. This is a direct consequence of the fact that Huffman coding is only optimal under certain circumstances. It was proved in paper [22] that the redundancy of Huffman coding, understood as the difference between the average Huffman codeword length and the entropy, is at most $p_{max}\;+\;0.086$, where $p_{max}$ is the probability of the most frequent symbol in the alphabet. In our example, this corresponds to the range between 1.385 and 1.995 bits per symbol, but the obtained result was close to the lower limit. The authors of [23] examined the extreme cases of large and small alphabets, and they concluded that, in the case of practical applications of text coding, the Huffman technique enabled obtaining codewords with an average length higher by less than 1% than the actual entropy. For shorter blocks of text (several hundreds of elements), the Huffman coding outperformed the arithmetic coding.

## 3. Proposed Method

The Huffman coding in its basic form allows the compression of data based on the frequencies of the occurrence of individual symbols from the alphabet. Hence, the lower limit for its performance can be calculated as the first-order entropy. This also means that even if there is redundancy in the input data due to the repetition of sequences of symbols, the basic Huffman coding is not able to use the information of these dependencies to compress the input data. In this case, in practical applications, we use dictionary-based methods that belong to the group of variable-to-block data compression techniques. It is also well known that variable-to-variable techniques may produce much better results in many cases than variable-to-block and block-to-variable approaches. For example, the Lempel–Ziv–Welch compression algorithm with a fixed length of the dictionary indexes will be easily outperformed by its variant where the length of the indexes increases with the size of the dictionary (see [12]). Combining run-length encoding (RLE) of data series with the Huffman technique can give very good results when certain symbols in the input data are repeated in long sequences. Another example is the DEFLATE algorithm [13], which combines the Huffman coding with the LZ77 dictionary algorithm, where the LZ77 is responsible for eliminating the repeated sequences of symbols, and the Huffman coding is responsible for replacing commonly used indexes with shorter binary representations.

In this paper, we focus on the variable-to-variable variant of the Huffman coding technique that operates on sequences of symbols, i.e., *m*-grams of input data symbols, where *m* can be several, several dozen, or even several hundred. This requires us to address the following issues: (i) a method of estimating the occurrence frequencies of the sequences of symbols in the input data, (ii) searching for the optimal Huffman coding algorithm in theory operating on *m*-grams (i.e., sequences of symbols). In particular, we need to estimate its computational complexity and practical applicability. From the practical point of view, an approximate and heuristic approach should be constructed, and (iii) the data decoding problem.

It should be noted that the problem of variable-to-variable Huffman coding has been addressed in the literature [15,16,17,18,19,20]. In papers [15,16,17,18], the authors proposed special variants of variable-to-variable Huffman coding techniques that were dedicated to the compression of data used in the process of testing systems-on-a-chip devices. Due to the specific field of application, the methods proposed there were adapted to the inherent attributes of the input data and thus cannot be used for general tasks. In particular, there were selective techniques that did not recover the values of bits unimportant in the testing process. The approach proposed in papers [19,20] was the dual-tree and variable-to-variable lossless entropy coding scheme. The first tree (Tunstall tree) was to parse the variable-length source words from the sequence of input data into indexes corresponding to single leaf nodes of the tree. The second tree (Huffman code tree) was used to map the indexes to variable-length codewords. By tuning both trees properly according to the probability distributions of words and code symbols, it enabled obtaining the asymptotic optimality of the scheme. However, it should be noted that the technique described in [19,20], different from our approach, had the following three typical properties and requirements: (i) It assumed that the input sequence was a random process with independent and identically distributed. random variables. (ii) The selection of the input sequence symbols according to the structure of the Tunstall tree had to be specified and unique. (iii) The asymptotic improvement of the efficiency of the Huffman coding depended on the combination of the two trees. It allowed obtaining the compression efficiency close only to the first-order entropy of input data. In our paper, we do not assume the statistical independence of the occurrences of the input data symbols. Furthermore, we do not use an encoding tree that uniquely defines the choice of the sequences of the input data symbols. In some sense, this increases the complexity of the task of optimal data encoding; however, it enables us to obtain the compression efficiency resulting from the *m*-gram entropy values (it should be noted that, by *m*-gram entropy, we understand entropy calculated for *m*-grams, i.e., *m*-element sequences of symbols, with m>1. For m=1, we can speak about the first-order entropy, which means the entropy calculated for individual symbols from the alphabet based on their frequencies of occurrence).

### 3.1. Estimating the Frequencies of the Symbol Sequences

The first step before creating the Huffman code tree is to estimate the frequency of the various symbol sequences that may appear in the input data. In our approach, we assume that *m*-grams or *m*-element sequences of symbols can form basic entities to which we assign succinct representations in the form of Huffman codewords. Therefore, first of all, we need to know the frequencies of their occurrences. Of course, *m*-grams and their frequencies can be set a priori, but in this paper, we consider an approach based on the so-called pattern input data (the pattern data is a specially selected set of data used to estimate the frequencies of the occurrence of *m*-grams, i.e., statistical characteristics of the data, which can be used further on to determine the Huffman codewords. It should be statistically representative for the given class of data), which allows us to find the sequences and their frequencies automatically using the proposed algorithm parametrized only with the maximum sequence length *m* (see Algorithm 1).
**Algorithm 1** Frequency estimation of the sequences of the input data elements**Require:**I, n=|I|, m≥1, n≥m, α≥0**Ensure:**$D_{F}$1:$D_{F}\leftarrow \left\{\right\}$                                ▹ Create an empty dictionary2:i←13:**while**i≤m**do**4:  j←05:  **while**
j≤n−i
**do**6:    S←I[j:j+i]                ▹ Select fragment of input data from *j* to j+i−17:    **if**
$D_{F}\left[S\right]=$**null**
**then**                    ▹ Fragment *S* is not in the dictionary8:      $D_{F}\left[S\right]\leftarrow i^{\alpha }$                   ▹ Add *S* to dictionary with initial frequency9:    **else**10:      $D_{F}\left[S\right]\leftarrow D_{F}\left[S\right]+i^{\alpha }$                      ▹ Increase frequency of *S*11:    **end if**12:    j←j+113:  **end while**14:  i←i+115:**end while**

The principle of operation of the algorithm is simple. It consists of two steps, i.e., iterating through a sequence I of the input data and extracting all possible sequences *S* of different lengths i=1,2,…,m. We also assume that we have a dictionary $D_{F}$ in which information about the frequencies of the occurrence of particular symbol sequences is stored. If the sequence *S* is already in the dictionary, we increase the number of its occurrences by the value of $i^{\alpha }$ (where *i* for i=1,2,…,m is the parameter indicating the size of the sequence of symbols under consideration, and α is the exponential scaling factor used to enhance the frequency of the occurrence of long sequences. We must note that with α=0 we count the actual numbers of the occurrences of the symbol sequences). Otherwise, we add *S* to the dictionary and initialize the occurrence counter of the sequence with a value equal to $i^{\alpha }$. The computational complexity of the algorithm can be roughly estimated as $O(m^{2}n)$.

We can see in Table 2 the results of the operation of Algorithm 1 for the following exemplary input data I={aaaaaaab}, where the maximum length of sequences m=3. It should be noted that the starting value of the counters, as well as their incremental values, depends on the α parameter. For α>0, the value of increment $i^{\alpha }$ is higher than 1 and depends on the size of the sequences. In this way, parameter α can be used to boost the frequencies of longer words. The third column of Table 2 shows the results obtained with α=1. This function is based on heuristics, but enables us to achieve better results in practical applications.

### 3.2. Determining the Huffman Codewords

Once we know the frequencies of the occurrence of *m*-grams for a given value of *m*, the next required step is to identify the Huffman codewords. This step requires creating the Huffman code tree using the standard procedure (described in Section 2). The only difference here lies in the fact that the leaf nodes hold not only single symbols from the alphabet A but also *i*-element sequences of symbols for i=1,2,…,m. In Table 3, we can see the Huffman codewords for our exemplary input data I for two different values of the α parameter.

It should be noted that the case of α=1 corresponds to the so-called weighted case. The values of the frequencies collected in this case are stored in the second column of Table 2. The assumed heuristic of weighing the frequencies of the occurrence of the symbol sequences allows us in this case to produce only one-bit long code for the most frequent sequence of symbols ‘aaa’. In Figure 4, we can see the Huffman code tree generated for our exemplary case and the weighted case with the α=1 variant of the frequency data.

In the case of high values of the parameter *m*, the number of the sequences of the input data can be too large, which makes it impossible to build the Huffman tree in an acceptable amount of time. To solve this problem, we propose the following heuristic, which keeps only a given percentage of the most frequent sequences of symbols of input data. At the same, it should be noted that we have to assign codewords to all individual symbols from the alphabet A in order to make the coding of the input data possible.

### 3.3. Optimal Coding Procedure

Once the codewords are generated and assigned to the sequences of symbols from the stream of input data, we can proceed to the next step of data coding. We must bear in mind, however, that in the considered case the data coding task differs from the similar task that must be performed in the case of the classical Huffman coding or the dual-tree coding approach introduced in papers [19,20]. The difference is that with both of the other techniques, there is always only one option of mapping a given sequence I of input data into the sequence of codewords C. For example, in the classical Huffman coding, we assign one codeword to each input symbol. Hence, for a given set of codewords, there is only one possible way to code a specific sequence of symbols. In the dual-tree approach, we use a Tunstall tree at each iteration of the algorithm in order to select the sequence of input data to which we assign an individual codeword. Once again this assignment is unique, and consequently, there is only one possible way to code a given set of input data. In the proposed approach, the ambiguity is fully expected. Let us consider the exemplary input data of the form I={aaaaaaab} and the codewords assigned to the sequences of symbols for m=3 (see Table 3). It is understood that we can start coding the input data in three different ways, that is by coding the ‘a’, ‘aa’ or ‘aaa’ sequence. Next, once again, we have the same three choices and so on until all the data are coded. Hence, we have several possible ways of coding the same input data, for example, (using codes for α=1):$$\begin{array}{ccc} \{\mathrm{aaaaaaab}\}\;=\;a+\mathrm{aaa}+\mathrm{aa}+\mathrm{ab} & \;\;\;\to \;\;\; & 111+0+10+11011\;=\;\{11101011011\},\\ \{\mathrm{aaaaaaab}\}\;=\;\mathrm{aaa}+\mathrm{aaa}+\mathrm{ab} & \;\;\;\to \;\;\; & 0+0+11011\;=\;\{0011011\},\\ \{\mathrm{aaaaaaab}\}\;=\;\mathrm{aa}+\mathrm{aaa}+\mathrm{aab} & \;\;\;\to \;\;\; & 10+0+1100\;=\;\{1001100\}. \end{array}$$
It is clear that in practical applications we are not interested in any other coding of input data apart from the coding that may guarantee the smallest number of bits. This way of data coding is called optimal. Although the whole problem is strictly combinatorial, it can be accelerated with the use of dynamic programming. In Algorithm 2, we can find the dedicated procedure allowing us to find the optimal coding C of the input data I based on the given set of codewords collected in the $D_{C}$ dictionary.

The proposed algorithm allows us to find the optimal code C for a given sequence of input data I (see Algorithm 2). It consists in building a directed graph stage by stage, which describes all possible ways of coding input data. See Figure 5 for an illustration in which each column represents a stage. By stages, we mean the subsequent steps of the algorithm leading to finding the optimal solution, while states can be interpreted as positions (indexes of symbols) in the sequence of input data. The nodes of the graph are labeled with the numbers {0,1,…,n}, and they represent all possible states of the process at the following stages. In fact, the nodes with the same number *n* correspond to one state *n*. Examining the graph, looking for a path leading from node 0 to node *n* is equivalent to analyzing the input data from the first symbol (state 0) to the end of the entire sequence (state *n*). Depending on stage *e*, where e=0,1,…,n, the set $S_{e}$ of possible states that we can proceed to in our next step is defined as:(2)$$\begin{array}{c} S_{e}\;=\;\left\{i:e\leq i\leq \min\left\{m\cdot e,n\right\}\right\}. \end{array}$$
It should be noted that being at stage *e* in state *i* we can change the state by moving to the next stage but only to one of the following possible states *j*, where the range of *j* is from i+1 to min{i+m,n}. The possible transitions between states are described with directed edges in the graph, which are also labeled with the sequences of the symbols from the input data by which we move forward in the whole input sequence. Hence, the resulting graph is a multistage graph, i.e., one that has only local connections between adjacent stages. Moreover, as a result, we can say that the entire process of searching for the optimal code has the Markov property, which means that the states at a given stage depend directly on the states and decisions made at the immediately preceding stage. The next step on the way to finding the optimal code (the solution to the problem) is to formulate the rule allowing us to find locally optimal solutions based on the previous results, i.e., the analogous results determined at the previous stage (the solutions to the subproblems).
**Algorithm 2** Procedure for finding the optimal coding of the input data**Require:**I, $D_{C}$, n=|I|, m≥1, n≥m**Ensure:**C1:$D_{nodes}\leftarrow \{0:\;$**null**}                     ▹ Create and initialize node data dictionary2:**for**e:=0**to**(n−1)**do**                   ▹ For loop with *n* iterations regarding stages3:  **for**
i:=e**to**
min{m∗e,n}
**do**                ▹ Iterate through nodes at each stage4:    k←i+1                     ▹ Calculate node indexes range in the next stage5:    l←i+m6:    **if**
l>n
**then**                     ▹ Node index cannot be higher than data size7:      l←n8:    **end if**9:    **for**
j:=k
**to**
l
**do**                         ▹ Iterate through node indexes10:      S←I[i:j]                    ▹ Read data sequence between *i* and j−111:      $c\leftarrow D_{nodes}\left[i\right]+D_{C}\left[S\right]$                    ▹ Calculate code sequence12:      **if**
$D_{nodes}$[*j*]=**null**
**then**                  ▹ Check if node *j* was already reached13:        $D_{nodes}\left[j\right]\leftarrow c$                  ▹ If not add it to $D_{nodes}$ with its code14:      **else**15:        **if**
$|D_{nodes}\left[j\right]|>|c|$
**then**                 ▹ Is new code sequence better?16:          $D_{nodes}\left[j\right]\leftarrow c$                 ▹ If needed update code sequence for *j*17:        **end if**18:      **end if**19:    **end for**20:  **end for**21:**end for**22:$C\leftarrow D_{nodes}\left[n\right]$                          ▹ Read the optimal coding sequence

Let us assume that $f_{e}\left(j\right)$ describes the length of the shortest path leading from node 0 at stage 0 to the current stage *e* (where e=0,1,…,n) and state $j\;\in \;S_{e}$. By the length of the path, we understand the total number of bits resulting from the input data partition along the path. This depends on the sequences of symbols along the path and the lengths of the codewords assigned to those sequences. Thanks to the Markov property of the problem, it is not difficult to find the shortest paths from the starting node 0 to all nodes at the current stage *e*, i.e., the values of $f_{e}\left(j\right)$ for $j\;\in \;S_{e}$. Specifically, for each node *j* at the present stage *e*, it is enough to check only the local connections to it leading from the nodes in the previous stage. We compute the length of the paths, which is certainly the summation of $f_{e-1}\left(i\right)$ and the length of the local connections, where $f_{e-1}\left(i\right)$ ($i\;\in \;S_{e-1}$) denotes the length of the shortest path from 0 to state *i* in stage e−1. It should be noted that $f_{0}\left(0\right)\;=\;0$. Then, we assign the minimal value to $f_{e}\left(j\right)$. Hence, following the Markov property we can write the following formula:(3)$$\begin{array}{c} f_{e}\left(j\right)\;=\;\;\;\;\;\;\;\min_{\begin{array}{c} i\in S_{e-1}\\ 1\leq j-i\leq m \end{array}}\;\;\;\;\;\;\;\left\{f_{e-1}\left(i\right)+\left|D_{C}\left[I\left[i\;:\;j\right]\right]\right|\right\} \end{array}$$
for $j\;\in \;S_{e}$, where $\left|\cdot \right|$ is the length of the code, and I[i:j] describes the sequence of the elements from the input data starting from index *i* to index j−1. Of course, in addition to the lengths $f_{e}\left(i\right)$ of the shortest paths, we also need to store the nodes that make up the paths (see Algorithm 2). In this way, we are able to find the sequence of codes corresponding to a given path. Consequently, we apply this rule to all the nodes at each stage until we obtain the final solution after reaching all nodes corresponding to the last state *n*. Since such nodes can be reached from more than one stage (actually from every stage starting with $e\;=\;\left\lfloor \frac{n}{m}\right\rfloor \;+\;1$), here, we have to choose the shortest path among all the possibilities. Such a path and the corresponding sequence of codewords is the final solution to the problem. The operation of finding the final solution to the problem is described in the graph representation (see Figure 5) in the form of a node labeled as n+1, which is connected to all the adequate nodes representing the state *n* at stages $e\;\geq \;\left\lfloor \frac{n}{m}\right\rfloor \;+\;1$.

In Figure 5, we can see the multistage graph created for the exemplary input data I={aaaaaaab}. We start to solve the problem of finding the optimal code with node 0 and stage e=0. It is obvious that the distance from node 0 to itself equals 0 (i.e., $f_{0}\left(0\right)\;=\;0$). It should be noted that in our example, we assumed m=3, which results in the possible values {1,2,3} of the states reachable for the next stage e=1 (i.e., $S_{1}\;=\;\left\{1,2,3\right\}$). Moving from node 0 at stage e=0 to nodes {1,2,3} at stage e=1 corresponds to coding the following sequences of symbols {a,aa,aaa} with the use of the codewords {111,10,0}, respectively. It gives the lengths {3,2,1} of the paths leading from the starting node 0 to nodes {1,2,3} at stage e=1. They are automatically the shortest paths from the starting node to the nodes considered at this stage (i.e., we have $f_{1}\left(1\right)\;=\;3$, $f_{1}\left(2\right)\;=\;2$, and $f_{1}\left(3\right)\;=\;1$). In the next step, we move to stage e=2. With m=3 and having the nodes {1,2,3} at the stage e=1, we can move from node 1 to nodes {2,3,4}, from node 2 to nodes {3,4,5}, and from node 3 to {4,5,6} (this means $S_{2}\;=\;\left\{2,3,4,5,6\right\}$). It corresponds to coding the following sequences of symbols taken from the input data: (i) moving from 1 to {2,3,4} means {a,aa,aaa}, (ii) moving from 2 to {3,4,5} takes {a,aa,aaa}, and (iii) moving from 3 to {4,5,6} corresponds to {a,aa,aaa}. All these sequences {a,aa,aaa} can be coded using the following codewords {111,10,0} with lengths {3,2,1}. Finding the shortest path for node 2 at stage e=2 is trivial because we have only one path from the nodes at stage e=1 to that node. It corresponds to the sequence ‘a’ from node 1 and codeword 111. Because the length of the shortest path for node 1 at stage e=1 is 3, we can say that the shortest length of the searched-for path is 6 (i.e., $f_{2}\left(2\right)\;=\;\max\left\{f_{1}\left(1\right)\;+\;3\right\}\;=\;6$). In the case of node 3, it is more complicated because it can be reached from two nodes {1,2} at stage e=1 with sequences {aa,aaa} and codewords {10,0}. It should be noted that the lengths of the shortest paths to nodes {1,2} at stage e=1 equal 3 and 2, respectively. Hence, in order to find the shortest path leading from the starting node to node 3 at stage e=2, we have to choose the smaller value from the sums 3+2=5 and 2+3=5 (which corresponds to $f_{2}\left(3\right)\;=\;\max\left\{f_{1}\left(1\right)\;+\;2,f_{1}\left(2\right)\;+\;3\right\}\;=\;\max\left\{5,5\right\}\;=\;5$). Both sums are equal so we choose the first possibility, which means the path moves from 0 to 1 and then to 3. We have to solve the problem of finding the shortest paths for the remaining nodes and the subsequent stages in the same way. It should be noted that the shortest paths for each node at every stage are indicated with by purple in the graph in Figure 5. In the case of node 8 (the final node), we have several shortest paths found because this node can be reached from several stages, which gives several candidates for the optimal path. Hence, the final solution to the problem is the shortest path selected among them, which in the graph is indicated by the additional node 9.

The computational complexity of Algorithm 2 can be roughly estimated as $O(m^{2}n^{2})$. It results from the number of *n* stages which we have to consider. Moreover, at each stage we have a number of nodes dependent on the size of input data *n*. Next, for each node we explore all possible connections to the nodes at the following stage in a number dependent on parameter *m*. Finally, in order to calculate the size of the code sequence in line 15 of the algorithm, we need to find the codeword for a sequence of data *S* taking advantage of dictionary $D_{C}$. This process also requires a number of operations relying on the value of parameter *m*. However, this complexity can be improved and reduced to $O(mn^{2})$ with the aid of the apt usage of the TRIE tree (shown for the considered exemplary input data in Figure 6). This requires combining the steps of generating the nodes for the next stage and selecting the sequences of the symbols corresponding to the transitions between stages, together with the process of finding the codewords assigned to them (this process can be accelerated with the use of the TRIE tree). In such a configuration, this process can be reduced to a number of operations of order O(m).

However, even for small values of parameter *m* the resulting computational complexity of the optimal algorithm is still proportional to the squared size of input data *n*. We realize that such computational complexity can be prohibitive in many practical applications, especially when operating on long sequences of input data.

### 3.4. Approximate Coding Procedure

In order to reduce the computational complexity of the coding process, we propose in this paper a more efficient but approximate approach, which is based on the greedy heuristic inspired by the solution to the well-known backpack problem [24]. The idea behind the proposed heuristic is simple and intuitive.

Let us assume that we are coding the given input sequence of data I and that we are currently at state *i* in stage *e*. We are going to move to the next stage e+1 by means of changing the state to *j*. For a given *m*, *j* could be any integer varying from j=i+1 to j=i+m, where j≤n. Clearly, every value represents a possible sequence to be coded in the next step, and our task is to choose a value of *j*. As opposed to the optimal one, in the heuristic approach, we may make the decision instantly while browsing the possible states at the next stage. The choice is based on the following rule: we choose the sequence of input data that guarantees the highest ratio of the length of the data sequence and the length of the codeword assigned to it. After choosing the best possible sequence, we add its codeword to C and then move to the next stage to the state resulting from the length of the chosen sequence. We continue these steps until the final node *n* is reached. The proposed heuristic approach is described in the form of pseudocode in Algorithm 3.

The computational complexity of Algorithm 3 lies between two limiting values. The first one is related to the case when all the sequences with lengths i=1,2,…,m are analyzed; meanwhile, the most beneficial is always the choice of a one element sequence. In this case, the computational complexity can be estimated as O(mn). The other case, also limiting, takes place when at each iteration of Algorithm 3 the most beneficial is the choice of the longest sequence. In this case, the computational complexity is of order O(n). With m≪n, we can say that the computational complexity of the approximate procedure described by Algorithm 3 is linear.

Let us return to the considered example of the input data I={aaaaaaab}. In order to test the effectiveness of both the optimal and approximate approaches for the two sets of codewords presented in Table 3, we provide the results of the I compression in the following part of this section.

For the optimal approach and the value of α=0, we obtain the following result C={1001000}, where |C|=7.With the approximate algorithm for α=0, we can obtain the following coding sequence C={0101110010} with length |C|=10.For the optimal approach and the value of α=1, we obtain C={1001100}, and the size of the coding sequence is |C|=7.With the approximate algorithm for α=1, it is possible to obtain the result in the form C={0011011} with the size of output data |C|=7.It should be noted that in the case of the classical Huffman coding we obtain codewords in the form of one-bit values 0 and 1, which are assigned to symbols ‘a’ and ‘b’; hence, the coding of the input data takes the form C={00000001} and |C|=8.

Based on the analysis of the above results, we can conclude that the proposed approach allows us to obtain better results in terms of the size of the output data than that resulting from the use of the classical Huffman coding (for a dictionary of size |A|=2, no compression can be achieved). In the optimal approach, regardless of the value of the α parameter, the length of the coded data was 7 bits. In the approximated approach, only in the second case, for α=1, was it possible to obtain a good result, equivalent to the optimal approach. This means that the proposed heuristic for boosting the frequency of the occurrence of the data sequences, fulfills its role. In Figure 7, we can see the multistage graph representing possible solutions of the given problem for the input data I={aaaaaaab} and α=1, where we mark the paths: (i) green for the optimal solution, (ii) red for the solution found with use of the approximate algorithm, and (iii) for illustration purposes, the path resulting from using the classical Huffman coding is blue.
**Algorithm 3** Approximate coding procedure**Require:**I, $D_{C}$, n=|I|, m≥1, n≥m**Ensure:**C1:C←{}                            ▹ Create and initialize output data2:i←0                            ▹ Initialize input data analysis index3:**while**i<n**do**4:  r←0                           ▹ Initialize best ratio found value5:  l←0                          ▹ Initialize end of best ratio sequence6:  **for**
k:=1
**to**
m
**do**                           ▹ Iterate from 1 to *m*7:    j←i+k                     ▹ Variable indicating end of the sequence8:    **if**
j>n
**then**                       ▹ Variable *j* cannot be higher than *n*9:      j←n10:    **end if**11:    S←I[i:j]                ▹ Read input data sequence between *i* and j−112:    $c\leftarrow D_{C}\left[S\right]$                    ▹ Obtain its codeword from $D_{C}$ dictionary13:    t←|S|/|c|                  ▹ Calculate the ratio and store it in variable *t*14:    **if**
t>r
**then**                    ▹ Better ratio found than previous values15:      r←t                           ▹ Change value of *r* variable16:      l←j                ▹ Remember value of the end index of the sequence17:    **end if**18:  **end for**19:  $C\leftarrow C+D_{C}\left[I\left[i:l\right]\right]$             ▹ Add to C the codeword for the best sequence20:  i←l                ▹ Change *i* according to the size of the best sequence found21:**end while**

### 3.5. Decoding Procedure

Since the proposed Huffman tree produces no ambiguity, the data decoding procedure follows the classical scheme, which is also used in the case of the classical Huffman decoding algorithm. Having the Huffman code tree, regardless of whether we have single symbols or their sequences at the leaf nodes, the decoding procedure consists of two steps, i.e., reading the consecutive bits from the coded data C and then finding the symbols by traversing the tree from the root node to the appropriate leaf nodes according to the directions resulting from the values of the bits read.

## 4. Experimental Results

In order to verify the effectiveness of the proposed approach, we conducted a series of experiments in lossless data compression for different values of the parameter *m*. The experiments involved the following types of data:Trajectory data: artificially generated data modeling the relative representation of the movement trajectories of objects;English texts: examples of natural spoken language taken from arbitrarily selected specimens of classical English prose;Fibonacci data sequence;DNA data;Floating point numbers representing measurement data;An executable object file.

The results obtained with the proposed approach were compared to the results obtained with the use of the variable-to-variable DEFLATE (DFLT) coding algorithm and the Prediction by Partial Matching (PPM) method. For the results see Table 4, Table 5, Table 6, Table 7, Table 8 and Table 9. It should be emphasized that, in our experiments, we assumed that for the proposed method the frequencies of the occurrence of the data symbols were evaluated on the basis of the pattern data, and they were used further on to determine the Huffman codewords. Moreover, it was also assumed that Huffman codewords were known for both the encoder and the decoder; hence, they were not stored in the output files obtained after the coding process.

### 4.1. Experiment 1

In this experiment we used artificially generated trajectory data, which modeled relatively described paths of object movements. The problem of trajectory coding is well known, and it concerns the task of the lossless compression of the movement data of various objects such as robots, drones, and vehicles, as well as hiking paths and contour data for cartographic purposes (see [25,26]). In order to code a path, we used the symbols A={a,b,c,d,e} drawn randomly with the following probabilities {0.88,0.05,0.01,0.05,0.01}. Then, an exemplary path description was as follows:


aaaaaaaeaaaaabaaaabaaaaaaaaaaacaaeaaaaaaaaeaabeaaaabaaaaaabaaaa



acaaeaaabaaaeaaaabaaaaaeaaeaaaaaaaaaaaaaaabaaeaaaaaeaaaaeaaaaaa



aebaabaaaaaaaaabaaaaaaaaeaaaaaeaaabaaaaaaaaeaaaaaaeeaaeaaaeaaaa


We must bear in mind that the movement paths were described in a relative way, which means that we had the initial absolute movement direction (north, north-east, east, etc.), and the following steps were described relative to the current orientation using the symbols from the alphabet A. The meaning of the symbols was as follows: ‘a’—continue moving in the same direction, ‘b’ and ‘d’—move slightly left or right, respectively, and ‘c’ or ‘e’—move left or right, respectively.

The results obtained in this experiment and expressed in the form of the entropy calculated for the different values of the length of the input data sequences (*m*-grams) are collected in Table 4. The efficiency in the data compression of the approximate variant of the proposed method (see Algorithm 3) was compared to the well-known variable-to-variable DEFLATE coding technique and the PPM algorithm.

**Table 4 entropy-24-01447-t004:** Results in compression of trajectory data.

*m*	m=1	m=2	m=4	m=6	m=8	DFLT	PPM
*H* [b]	1.20	0.89	0.82	0.85	0.88	1.01	0.841

On the basis of the analysis of the results from Table 4, we can conclude that in this experiment, the proposed method allowed obtaining much better results than the well-known DEFLATE coding technique and also better results than those obtained with the use of the PPM prediction technique. Moreover, the smallest values of entropy were obtained for m=4.

In the second part of this experiment, we searched for the answer to the question: how close are the results obtained with the approximate algorithm to the optimal results? For this purpose, we conducted a series of subexperiments operating on short data sequences (due to the high computational complexity of the optimal approach) for both Algorithms 2 and 3 and compared the obtained results (see Table 5).

**Table 5 entropy-24-01447-t005:** Comparison of the compression efficiency of the optimal and approximate approaches.

*m*	m=1	m=2	m=4	m=6	m=8
Data size (optimal) [B]	152	114	105	107	112
Data size (approx.) [B]	152	114	107	108	115

Based on the results obtained in this experiment, we can conclude that the approximate approach gave results identical to (for m=2) or very close to those obtained with the optimal approach. Here, the difference is only about 1% to 3% of the resulting value.

### 4.2. Experiment 2

The aim of the second experiment was to verify the effectiveness of the proposed approximate algorithm of the *m*-gram entropy Huffman coding with application to the lossless compression of texts written in English. In order to do this, we carried out the experiment with the use of arbitrarily selected texts written in English, where the number of alphanumeric characters was reduced to the set including a blank space and lowercase letters ‘a’ to ‘z’. The obtained results are collected in Table 6.

**Table 6 entropy-24-01447-t006:** Entropy of English language measured for the growing sizes of *m*-grams operating on exemplary texts.

*m*	m=1	m=2	m=4	m=6	m=8	DFLT	PPM
*H* [b]	4.11	3.87	3.21	2.85	2.83	2.89	2.12

Based on the analysis of the received values of the entropy, we observe that the proposed approximate approach obtained better results than the DEFLATE technique. Moreover, the outcomes of the experiment were consistent with the results obtained, for example, by Claude E. Shannon for *m*-grams (see e.g., [27]), where the entropy of the English language for m=1 was estimated as 4.16 bits, and in the case of *m*-grams, i.e., operating on whole words, the result was around 2.23 bits. However, the best results were obtained with the use of the prediction by partial matching technique, which enabled construction online of the best statistics for this specific data.

### 4.3. Experiment 3

The binary Fibonacci sequence is a sequence of symbols generated over the two-symbol alphabet A={a,b} according to the following rule, which is applied recursively. In the first place, we start with the initial sequence $I_{1}\;=\;\left\{a,b\right\}$. Next, and also in the following iterations, the sequence obtained in the previous step is analyzed, and the individual occurrences of ‘a’ or ‘ba’ are replaced by sequence {a,b}, and the symbol ‘b’ is replaced by ‘ba’. For example, after the first four iterations of the algorithm we have: $I_{1}\;=\;\left\{a,b\right\}$, $I_{2}\;=\;\left\{a,b,\mathrm{ba}\right\}$, $I_{3}\;=\;\left\{a,b,\mathrm{ba},a,b\right\}$, and $I_{4}\;=\;\left\{a,b,\mathrm{ba},a,b,a,b,\mathrm{ba}\right\}$. After combining the elements of the resulting data, we obtain a binary Fibonacci sequence of data, e.g.,:


abbaababbaabbaababbaababbaabbaababbaabbaababbaababbaabbaababbaa



babbaabbaababbaabbaababbaababbaabbaababbaabbaababbaababbaabbaab



abbaababbaabbaababbaabbaababbaababbaabbaababbaababbaabbaababbaa


It should be noted that the frequencies of the occurrence of both symbols in the binary Fibonacci sequence are identical, and the lengths of the sequences obtained at the following iterations of the generation algorithm are the consecutive Fibonacci numbers. In this case, i.e., when the frequencies of the symbols are identical, the classical Huffman coding (with m=1) does not allow any compression. The results obtained in this experiment for different values of the parameter *m* and the approximate technique (see Algorithm 3) are collected in Table 7.

**Table 7 entropy-24-01447-t007:** Results of the compression of the Fibonacci data sequence.

*m*	m=1	m=16	m=64	m=256	m=1024	DFLT	PPM
*H* [b]	1.0	0.3	0.1	0.044	0.032	0.035	0.093

Based on the analysis of the results obtained in this experiment, we can draw the following conclusions: (i) the classical Huffman coding (m=1) does not allow for data compression, (ii) the proposed approach allows us to obtain a significant reduction in the size of data, (iii) the results received with the proposed approximate *m*-gram entropy Huffman coding scheme were better than the results from the DEFLATE coding for high values of *m*. It should be noted that in this experiment, it was possible to receive results better than the use of the DEFLATE technique only for high sizes of *m*-grams, i.e., with m=1024. This is due to the specifics of the data, where long sequences of data are repeated. In order to make the process of Huffman code generation possible, in this experiment, we used the previously mentioned heuristic that consists in taking into account only a given percentage of the most frequently repeated sequences of data. In this experiment, it was equal to 1%. The worst results were obtained with the use of the PPM technique even with manually setting to 1024 the number of symbols taken into consideration during the prediction process.

### 4.4. Experiment 4

The following experiment involved DNA sequencing data over the alphabet A={A,C,T,G} representing four possible nitrogen bases. An exemplary fragment of the input data is shown below:


AGCTTTTCATTCTGACTGCAACGGGCAATATGTCTCTGTGTGGATTAAAAAAAGAGTGTCTGA



TAGCAGCTTCTGAACTGGTTACCTGCCGTGAGTAAATTAAAATTTTATTGACTTAGGTCACTA



AATACTTTAACCAATATAGGCATAGCGCACAGACAGATAAAAATTACAGAGTACACAACATCC



ATGAAACGCATTAGCACCACCATTACCACCACCATCACCATTACCACAGGTAACGGTGCGGGC


The results of the compression using the proposed approximate approach (see Algorithm 3) and both the DEFLATE and PPM techniques are collected in Table 8.

**Table 8 entropy-24-01447-t008:** Results of the compression of the DNA sequencing data.

*m*	m=1	m=2	m=4	m=6	m=8	DFLT	PPM
*H* [b]	2.00	2.02	1.98	1.96	1.95	2.17	2.11

Due to the specificity of the input data, only a slight reduction in data size was possible (the best results were obtained for m=8). However, the results obtained with the proposed approximate approach were better than results received with the aid of the DEFLATE technique and the PPM algorithm.

In experiments 1–4, the input data was divided into pattern and test sets, both of sizes around $10^{6}$ symbols. The pattern sets were used to find the frequency of the occurrence of the *m*-grams and to determine the code words, which constitute the necessary a priori knowledge for the proposed method on this basis. In turn, the test sets were used to assess the efficiency of the proposed method on the data compression.

### 4.5. Experiment 5

In this experiment, we verify the effectiveness of the proposed approach and compare it with the results obtained for the DEFLATE and PPM algorithms, operating on the popular Calgary Corpus data set. In Table 9, we show the results obtained for the representative sets of data belonging to the following classes: floating-point numbers (GEO), English text (BOOK1, BIB), and binary executable data (OBJ1). It should be noted that in the case of all the considered classes of data, the pattern and test sets were selected from the input data in the proportions 25% to 75%. This enabled finding the necessary a priori knowledge for the method in the form of the frequencies of the occurrence of the sequences of the symbols and the code words assigned to them. Moreover, the GEO and OBJ1 data sets were analyzed over the binary alphabet, and BOOK1 and BIB were analyzed over the alphabet of alphanumeric symbols.

**Table 9 entropy-24-01447-t009:** Results of the entropy *H*[b] for the Calgary Corpus dataset.

GEO							
m=2	m=4	m=8	m=12	m=14	m=16	**DFLT**	**PPM**
0.923	0.880	0.828	0.816	0.800	0.786	1.286	0.823
**BOOK1**							
m=1	m=2	m=4	m=6	m=8	−−	**DFLT**	**PPM**
4.134	3.805	3.255	3.365	3.929	−−	4.144	2.228
**BIB**							
m=1	m=2	m=4	m=6	m=8	−−	**DFLT**	**PPM**
5.225	4.462	3.681	3.502	3.610	−−	2.694	1.918
**OBJ1**							
m=2	m=4	m=8	m=12	m=14	m=16	**DFLT**	**PPM**
1.023	0.924	0.893	0.853	0.826	0.789	0.706	0.834

The proposed method obtained the best results only in the case of GEO data. In the case of the OBJ1 data set, the obtained results were close to but worse than the results obtained with the use of the DEFLATE algorithm and much better than the results obtained with PPM technique. In the case of the text data, the PPM method obtained the best results, and the proposed method performed the worst. This was caused by the imprecise estimation of the frequency of the occurrence of the symbol sequences on the basis of the pattern sets.

On the basis of the obtained results, we can conclude that the proposed approach provides good results in the case of data sets characterized by constant and relatively easy to determine statistics of the occurrence of a sequence of symbols. Examples of such data are: trajectory data, Fibonacci sequences, DNA sequencing data, or the sets of floating point numbers. In the case of the text data, where such an estimate is much more difficult to perform, the proposed method had worse results. It is worth showing the results of the experiment for the BIB set, where the pattern set, i.e., the one used to determine the statistical characteristics of the data, was also coded. The results of this experiment were: 4.381 bps for m=2, 3.129 bps for m=4, 2.279 bps for m=6, and 1.775 for m=8. The last result was much better than the results obtained with the DEFLATE (3.094 bps) and PPM (2.470 bps) algorithms.

## 5. Conclusions

In this paper, we proposed optimal and approximate approaches to *m*-gram entropy variable-to-variable data coding, which are natural extensions of the classic Huffman coding algorithm. Due to the quadratic computational complexity $O(mn^{2})$, the optimal approach can be treated as a benchmark for approximate algorithms, or it can be used to code short sequences of input data. In turn, the proposed approximate approach is characterized by linear complexity O(mn), and as such, it can be used in the practical applications of lossless data compression. In order to verify the effectiveness of the proposed approximate approach, we conducted a series of experiments operating on different kinds of practical input data. On the basis of the obtained results, we can conclude that the proposed approximate approach gives good results for data characterized by invariant and easy to evaluate statistics of the occurrence of the sequences of the symbols. For example, for trajectory data or DNA sequencing data, it was possible to obtain results that were better by 2.5 and 7.6 percent, respectively, than the popular DEFLATE and PPM algorithms. In the case of the text, the results for the proposed method were worse than for the DEFLATE and PPM algorithms. On this basis, it can be concluded that the proposed method can find practical application in the compression of data having the aforementioned property. Moreover, in this paper we proposed a procedure for determining the frequencies of the occurrence of *m*-element sequences of symbols (*m*-grams) in a string of input data, and we also discussed the problem of data decoding. Possible directions for future work may include improvements to the optimal algorithm, including estimation of the frequencies of the occurrence of *m*-grams and also the development of more effective approximate techniques.

## Figures and Tables

**Figure 1 entropy-24-01447-f001:**
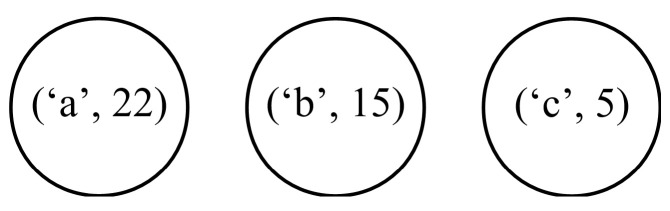
Initial forest of nodes representing the symbols from the alphabet together with their frequencies of occurrence.

**Figure 2 entropy-24-01447-f002:**
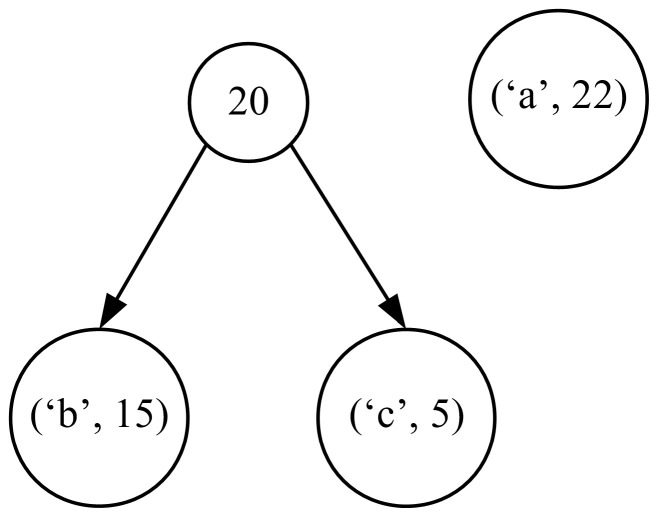
Operations within each iteration of the Huffman code’s tree-building procedure consisting in finding the two trees with the smallest total number of occurrences of symbols and combining them into a single tree.

**Figure 3 entropy-24-01447-f003:**
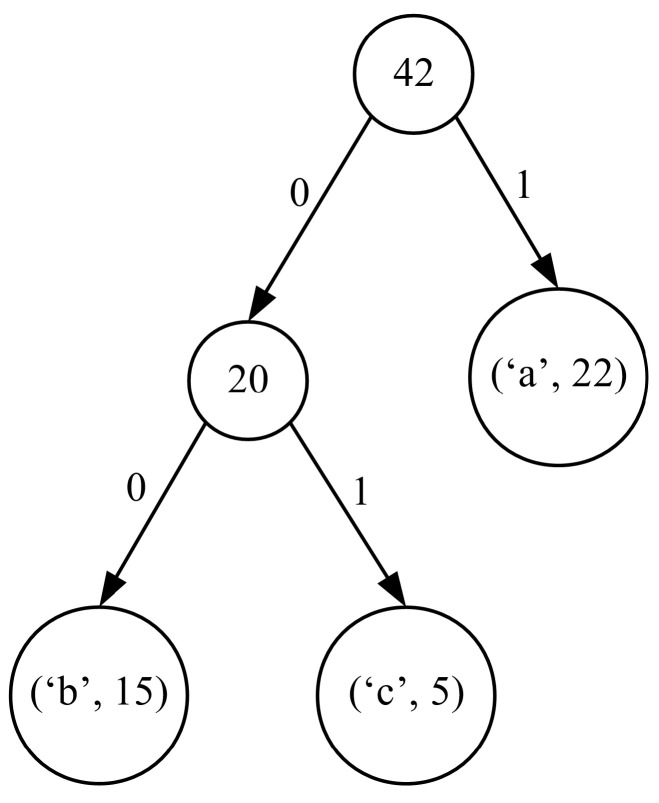
The Huffman code tree obtained for the considered exemplary case.

**Figure 4 entropy-24-01447-f004:**
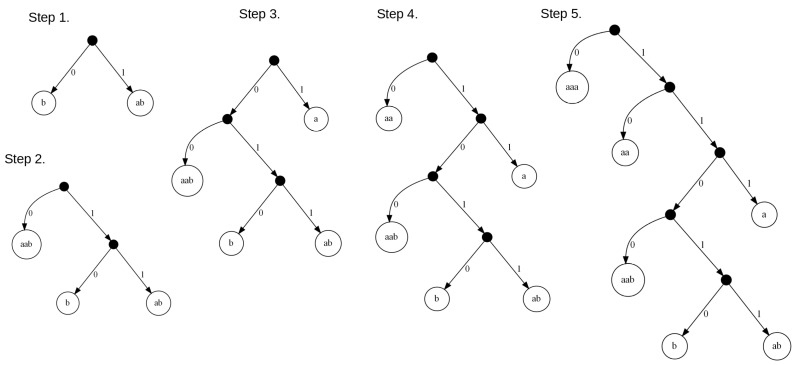
Steps required to build the Huffman code tree for the weighted variant of the frequency calculation algorithm in the case of the exemplary input data I.

**Figure 5 entropy-24-01447-f005:**
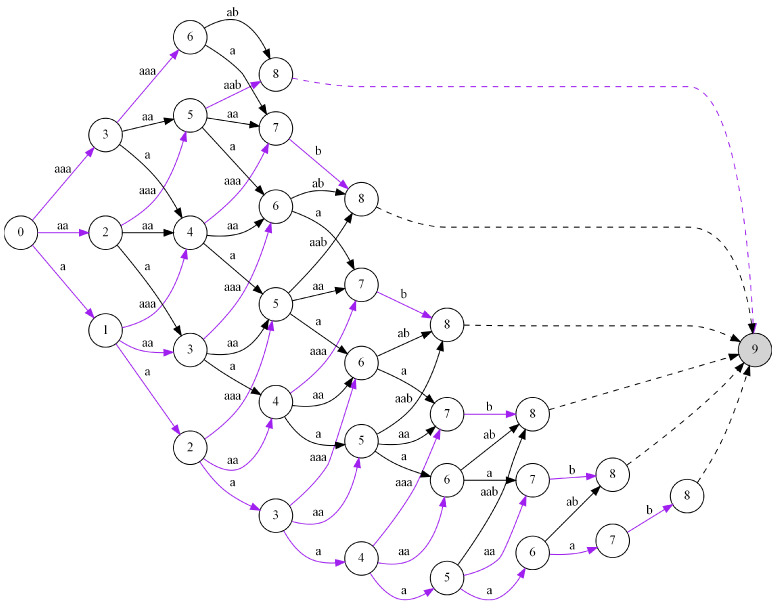
The multistage graph created for the exemplary input data I={aaaaaaab}.

**Figure 6 entropy-24-01447-f006:**
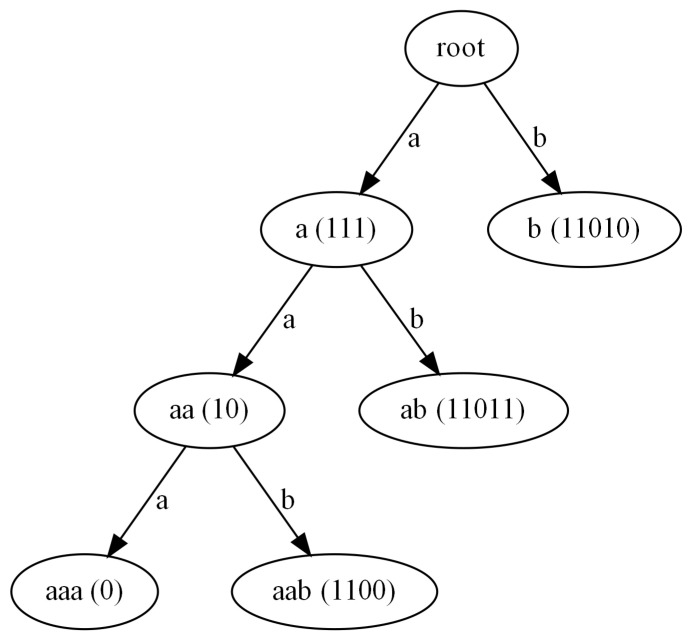
The TRIE tree generated for the exemplary input data and the codewords assigned to the sequences of symbols for m=3.

**Figure 7 entropy-24-01447-f007:**
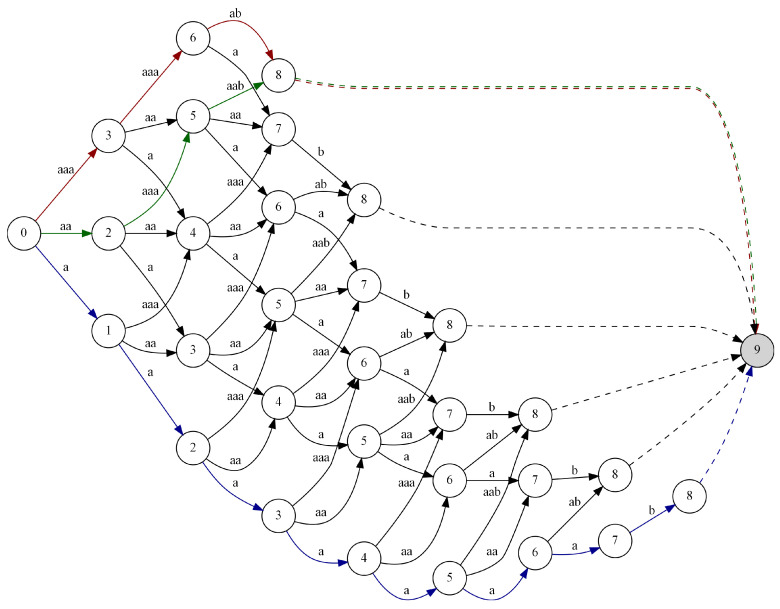
The multistage graph for the input data I={aaaaaaab} with the possible solutions obtained with α=1 marked.

**Table 1 entropy-24-01447-t001:** The frequency of the occurrence and the codewords assigned to the individual symbols coming from the exemplary input sequence I.

Symbol	Frequency	Codeword
a	22	1
b	15	00
c	5	01

**Table 2 entropy-24-01447-t002:** The frequencies and the weighted (α=1) frequencies of the occurrence of sequences in the input data I for m=3.

Symbols	Frequency	Weighted (α=1)
a	7	7
b	1	1
aa	6	12
ab	1	2
aaa	5	15
aab	1	3

**Table 3 entropy-24-01447-t003:** Huffman codewords determined for the sequences of symbols based on an exemplary message for m=3 and for two different values of parameter α.

Symbols	α=0	α=1
a	11	111
b	0010	11010
aa	10	10
ab	0011	11011
aaa	01	0
aab	000	1100

## Data Availability

Not applicable.

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
