# Peer review of "Variable-to-Variable Huffman Coding: Optimal and Greedy Approaches"

_entropy, 2022, doi:10.3390/e24101447_

Round 1

Reviewer 1 Report

In this paper, the authors propose a combination of Huffman-type coding with a technique based on constructing a dictionary of variable-length data sequences.  The paper is clearly written but there is a lack of some necessary references and comparisons. The main remarks are given below.

 1.      The authors state that their technique allows “to obtain compression efficiency resulting from the higher order entropy values”. I did not find any theoretical arguments confirming this statement.

2.   What means “training”  input data for constructing a dictionary with input symbols and their probabilities? Does it mean offline processing of the sequence to be encoded?  If so, what is new in this approach?  This issue should be clarified. For me, the suggested approach looks like two-pass encoding. Is it so?

3.   I would recommend to compare the suggested approach with the prediction by partial matching (PPM) and context tree weighting (CTW) algorithms. The corresponding reference should be included.

4.     In the data compression area there exist conventional databases for comparative studying of data compression techniques. Classical examples are Calgary Corpus and Canterbury Corpus. Comparison with published results for other data compression techniques is necessary.

Author Response

We would like to thank you for your valuable comments that allowed us to increase the quality of our work. Detailed references to individual comments can be found in the pdf file.

Reviewer 2 Report

Summary:

This paper proposes a new Huffman encoding algorithm that supports compression from multiple symbols to a code. The authors describes the algorithms of the frequency counting, the minimal tree creations. The experimental evaluations show better results comparing to the DEFLATE algorithm.

Comments:

- The approach and the strategy of the research is well considered. However, the manuscript has fatal drawback of readability. The English presentations should be drastically improved. Especially, there are many sentences  written in a single and very long expressions. For example, the following sentences are very long. It is not understandable. : Line 76-79, 158-161, 201-206, 207-212, 254-257, 311-315, 388-393, 442-445.

- In section 3, it is not clear what is "the first order entropy". Is the entropy when the Huffman coding with a single symbol is applied?

- It is not clear how the codes in Table 3 are derived. Before that, it is not clear what are the alpha and i. Because the definitions of those parameters are ambiguous, it is hard to understand the algorithm 2.

- It is hard to understand the lines 285-306. Please revise the expressions and English presentations.

- In the equation (2), what is "me"? Is it "m x e"?

- The authors are discussing the compression  performance only focusing on the coding sizes. Actually, the total compressed data size should include the data structure of the encoding graph (TRIE tree in the manuscript). The compression performance including the tree structure should be discussed in  the evaluation section.

Author Response

(The authors gave the same response as above.)

Reviewer 3 Report

This is a generally well-written paper, on the topic of lossless data compression, where the idea is to design an hybrid dictionary/variable-length coding method (variable-to-variable). The proposal looks interesting but lacks extensive validation. In fact, comparing only with deflate is clearly insufficient. Many other methods are available and should be used for comparison, both regarding compression ratios and computational complexity. See, for example, http://mattmahoney.net/dc/

Also, you should use the compression corpus that are traditionally used for benchmarking the algorithms (for example, the Calgary corpus is a small one, but there are others).

Author Response

(The authors gave the same response as above.)

Round 2

Reviewer 1 Report

After revision of the paper, it became more clear that:

1.      In fact, the authors consider a two-pass version of the ZLW method [10,11]. Variable length coding of the indices in the tables constructed by the ZLW algorithm has been mostly dealt with in patents as it is more of a practical implementation issue than a  scientific contribution.

2.      Comparison with the PPM algorithm shows (Table 9) that the PPM algorithm has almost 2 times better compression than the proposed method. Moreover, the comparison is not entirely correct  since

·        PPM is on-line (one-pass) coding

·        PPM is a universal method that does not require any a priori information about  the source

·        In the case of GEO, the data format is not taken into account by both the PPM and DFLT but the authors used this information for their algorithm. In my opinion, not mentioning these differences in the algorithms would be misleading for potential readers.  

Author Response

Dear Reviewer,

In the first place, we would like to thank you for all the comments which allowed us to improve the quality of our article.

1. In fact, the authors consider a two-pass version of the LZW method [10,11]. Variable length coding of the indices in the tables constructed by the LZW algorithm has been mostly dealt with in patents as it is more of a practical implementation issue than a scientific contribution.

We fully agree with the comments. The proposed approach is the two-pass one and it needs a priori information about input data statistics. In the first pass, which is supposed to be done once operating on the so called pattern data, we evaluate frequencies of occurrence of m-element sequences in input data where m is changing from 1 to the value given as the parameter of the method. This is the crucial step because the efficiency of the method relies on the accurate estimation of those frequencies. Hence, it is very important to choose the pattern data as the statistically representative for the given class of data. Because the information about the frequencies in the case of long sequences of data would take a lot of space it is impractical to store that information in the output file. That is way we assume this information to be known both by the coder and decoder. The similar approach was used in JPEG file where we could take advantage of default Huffman tables. To sum up the role of the first step is to estimate the frequencies of occurrence of sequences of symbols of lengths changing from 1 to the given value while operating over the alphabet defined by the data itself. This is the natural extension of Huffman coding from single symbols to sequences of symbols. Of course there is the similarity with LZW method where the dictionary is being filled online with longer and longer sequences of symbols. But the proposed approach is a direct extension of Huffman coding. Once the frequencies are estimated it is possible to assign to them code words based on the binary tree according to the heuristic: the more frequent the sequence – the shorter the code word assigned to it. Here, we propose the algorithm allowing to estimate the frequencies of occurrence of sequences of symbols. We also propose the heuristic allowing to emphasize longer sequences and in this way assign to them shorter codes (of course still in connection with the frequencies of their occurrence).

The process of data coding in the case of sequences of data taken from the input stream is also more complex than in the case of standard Huffman coding. In standard Huffman coding it was possible to assign code words to individual symbols and the coding process was unique. In the case of the proposed approach we have to take into consideration sequences of symbols and their all combinations so the optimal coding process seems to be combinatorial. However, in the paper we manage to prove that with the use of dynamic programming the optimal coding can be found with O(mn^2) time which we found to be an interesting theoretical result. Since that complexity (square) can be too high for practical usage we also propose the heuristic leading to the approximate approach where in the coding process we choose locally those sequences which are characterized by the most beneficial ratio of the sequence length to the length of the code word assigned to it. In our opinion, this is also a novel approach.

2. Comparison with the PPM algorithm shows (Table 9) that the PPM algorithm has almost 2 times better compression than the proposed method. Moreover, the comparison is not entirely correct since

· PPM is on-line (one-pass) coding

· PPM is a universal method that does not require any a priori information

about the source

We fully agree with the comment. The PPM algorithm, which belongs to the class of context coding, is very powerful and in the case of some of the considered types of data it allows to obtain better results. Here, in PPM, the context (frequencies of occurrence of sequences of symbols) is being built and updated online so in the case of long data it can be better suited to the input stream. Especially when we deal with data where the mentioned statistical characteristics are hard to be properly estimated a priori. In this way PPM does not require any a priori information. However, the proposed method seems to give better results in the case of data which can be characterized with invariable and possible to be correctly estimated the statistical characteristics (i.e. GEO, DNA, trajectory data). So here we prove the advantage of our method and the possible field of its application.

In our opinion the proposed method has higher potential than PPM in some cases which is understood in the following sense:

- PPM builds the context online which means that it can’t be optimally selected at least from the beginning of data stream. The proposed method, if only it was possible to estimate the frequencies of occurrence of sequences of symbols correctly, starts with ‘optimal’ characteristics from the very beginning. Of course we assume here a priori knowledge;

- PPM in the coding process uses different heuristic which means that the longest context is considered first. Longest contexts doesn’t have to be optimal. In our approach we use, in our opinion, better suited heuristic which takes into consideration the mentioned ratio of the sequence length to the length of the code words assigned to it;

- we proposed the optimal coding algorithm. Of course we agree that its computational complexity can be prohibitive in the case of long data to be coded but as long as we deal with short streams it can be used in practical applications.

Of course PPM is a universal algorithm and it needs no a priori knowledge. This make our method different and specifies its possible fields of applications.

3. In the case of GEO, the data format is not taken into account by both the PPM and DFLT but the authors used this information for their algorithm. In my opinion, not mentioning these differences in the algorithms would be misleading for potential readers.

Of course we agree with the comment. In the case of GEO data we were operating on its binary representation and the only a priori knowledge were the frequencies of occurrence of sequences of bits taken from the input data which is the assumed by the method a priori knowledge (the knowledge that must be known by the method in the similar way like in the case of standard Huffman or arithmetic coding but here extended to the sequences of symbols).

Best regards,

Authors

Reviewer 2 Report

Authors have responded to all negative comments and they revised the parts in the manuscript appropriately.

However, there are still hardness in readability regarding English presentations. It is better to ask some English native person to proofread the manuscript.

Author Response

Dear Reviewer, 

In the first place, we would like to thank you for all the comments which allowed us to improve the quality of our article.

We fully agree with the comments. We reread the paper and found many sentences which were too long and needed to be changed. All the changes are indicated in the paper with brown color. We also managed to correct a couple of syntax errors and typos. We believe that in this way the readability of our paper was highly improved.

Best regards,
Authors

Reviewer 3 Report

The authors have provided enough relevant additional content in this revised version, hence, my opinion is that it is ready to be published.

Author Response

Dear Reviewer,

We would like to thank you once more for all your valuable comments that allowed us to improve the quality of our article.

Best regards,
Authors